# Conductive Hydrogels Based on Industrial Lignin: Opportunities and Challenges

**DOI:** 10.3390/polym14183739

**Published:** 2022-09-07

**Authors:** Chao Liu, Yu Li, Jingshun Zhuang, Zhouyang Xiang, Weikun Jiang, Shuaiming He, Huining Xiao

**Affiliations:** 1Jiangsu Co-Innovation Center of Efficient Processing and Utilization of Forest Resources, Nanjing Forestry University, Nanjing 210037, China; 2State Key Laboratory of Pulp and Paper Engineering, South China University of Technology, Guangzhou 510640, China; 3State Key Laboratory of Biobased Material and Green Papermaking, Qilu University of Technology, Shandong Academy of Sciences, Jinan 250353, China; 4Department of Chemical Engineering, University of New Brunswick, Fredericton, NB E3B 5A3, Canada

**Keywords:** industrial lignin, hydrogel, conductivity, preparation, application

## Abstract

The development of green materials, especially the preparation of high-performance conductive hydrogels from biodegradable biomass materials, is of great importance and has received worldwide attention. As an aromatic polymer found in many natural biomass resources, lignin has the advantage of being renewable, biodegradable, non-toxic, widely available, and inexpensive. The unique physicochemical properties of lignin, such as the presence of hydroxyl, carboxyl, and sulfonate groups, make it promising for use in composite conductive hydrogels. In this review, the source, structure, and reaction characteristics of industrial lignin are provided. Description of the preparation method (physical and chemical strategies) of lignin-based conductive hydrogel is elaborated along with their several important properties, such as electrical conductivity, mechanical properties, and porous structure. Furthermore, we provide insights into the latest research advances in industrial lignin conductive hydrogels, including biosensors, strain sensors, flexible energy storage devices, and other emerging applications. Finally, the prospects and challenges for the development of lignin-conductive hydrogels are presented.

## 1. Introduction

Polymer hydrogel is a functional material with a three-dimensional network structure formed by crosslinking of hydrophilic polymers, which has the inherent ability to swell in water/aqueous solvents. Biomass-based hydrogels have good biocompatibility, degradability, low toxicity, unique flexible elastic mechanical properties, and swelling and extension characteristics [1]. In particular, new biomass-based conductive hydrogel materials with electrical response properties show broad application prospects in bioelectronic applications such as bioelectronic sensors, flexible electronic devices, supercapacitors, and tissue engineering [2,3]. The basic idea of designing conductive composite hydrogels is to realize the conductive properties of hydrogel materials by adding or introducing conductive particles (or fillers), conductive polymers, and conductive ions during the preparation of hydrogel substrates [4,5]. Although research on adding conductive materials to bio-based hydrogels is continuously reported, the incompatibility of flexibility and conductivity is still a bottleneck in the research field of bio-based conductive hydrogel materials.

To promote the practical application of hydrogels and improve their poor mechanical properties, construction strategies such as dual networks and interlocking crosslinked networks are proposed. Hydrogels with various three-dimensional network structures, such as topological structures and interpenetrating/semi-interpenetrating network structures, are synthesized [6,7]. Many research interests have shifted toward bio-based hydrogels. Biopolymers such as cellulose, lignin, chitin, amino acids, and proteins derived from biomass resources are widely used to prepare hydrogels due to their good biocompatibility and environmental friendliness [8,9]. As a sustainable, abundant, and aromatic polymer in lignocellulosic biomass, lignin is widely utilized as a building framework in hydrogels. The structure of lignin contains a large number of functional groups (such as alcoholic hydroxyl groups, phenolic hydroxyl groups, and carboxyl groups) and chemically reactive sites (such as C_3_ and C_5_ positions of the aromatic ring of lignin), which can be chemically modified and graft copolymerized to prepare lignin-based hydrogels [10,11].

Generally, lignin is mostly used as an electrode and electrolyte material. Compared with its research in the field of energy storage, the research of lignin in the field of flexible conductive hydrogel materials is still in its infancy. As is known, the common operating mechanism of hydrogel for flexible electronic devices is that external stimuli forces (such as strain or pressure) are transformed into detectable electrical signals (resistance, voltage, current, or capacitance) [12]. Thus, sufficient conductivity is a vital factor affecting the application of hydrogels in flexible electronic devices. Similarly, excellent adhesion is necessary for hydrogel-based wearable electronics, which prevents easy falling-off from the human body and enhances the sensitivity of signal identification [13]. In addition, wearable electronics demand ultra-strong stretchability to prevent breakage during human body movement [14]. Beyond that, hydrogel-based flexible electronics should have other features and functions, such as the ability of UV-resistance and antibacterial properties [15,16]. In fact, the properties of lignin, such as its role as an antioxidant and antimicrobial, its high-temperature resistance, UV absorption, abundant hydrophilic functional groups (OH groups), ability to retain moisture for a long time, and adsorption of selected chemicals, make it an ideal substitute for synthetic polymers in the production of hydrogels. Furthermore, the addition of lignin can flexibly alter the mechanical properties and permeability of the hydrogel [17,18].

In addition to being used in the above-mentioned flexible electronic devices, lignin-based conductive hydrogels are also widely used in metal/reagent detection, heavy metal ion adsorption, energy storage (lithium batteries, supercapacitors), and other fields [19,20]. However, current review articles on lignin-based conductive hydrogels are few and mainly focus on applications in flexible electronic devices. Few review articles summarize the applications of lignin-based conductive hydrogels in different fields. This mini-review aims to present a holistic discussion on lignin-based conductive hydrogels starting from the sources of industrial lignin, techniques for preparation of lignin-based hydrogel, properties of the lignin-based conductive hydrogel, and a comprehensive review on its application via bio-sensors, strain and pressure sensor devices, flexible energy storage, and others (Figure 1). Finally, some illustrations regarding the future perspectives of industrial lignin-based conductive hydrogel are presented.

## 2. Lignin from the Pulping Process

Lignin is a multi-branched, amorphous natural substance with a phenylpropane structure, which contains phenolic hydroxyl, an aromatic ring, carboxyl, alcoholic hydroxyl, methoxy, carbonyl, a conjugated double bond, and other structures [21]. The main structural units are as follows: (1) the coniferyl alcohol unit, (2) the sinapyl alcohol unit, and (3) the *p*-coumary alcohol unit [22], as shown in Figure 2a. These three basic structural units are generally combined by carbon–carbon bonds or ether bonds to form random molecules with a three-dimensional network structure.

Based on different production scales, lignin preparation methods can generally be divided into two categories: one is laboratory-produced lignin, and the other is industrially produced lignin, including lignin from the pulp and paper industry and biomass refineries. Industrial lignin was first realized by the pulping process [23], which is the lignin mainly mentioned in this review paper. According to different pulping processes, lignin is mainly divided into four different types, namely kraft lignin, lignosulfonate, soda lignin, and organosolv lignin (Figure 2b).

### 2.1. Kraft Lignin

Currently, most of the lignin in the pulping industry is generally extracted from kraft black liquor [24]. Basically, alkaline solution systems (such as sodium sulfide and sodium hydroxide solutions, etc.) are used in the kraft pulping process. Lignin depolymerization mainly occurs in the ether linkages connected at the α or β positions of the side chains of the lignin building blocks. The ether bond of the lignin polymer is broken and decomposed into small molecular chains, which are dissolved in an alkaline solution to form black liquor [23,25].

### 2.2. Lignosulfonate

Sulfite pulping uses a similar process to kraft pulping, but sulfite pulping is performed in an acidic solution medium [26]. Due to the introduction of a large number of sulfonate groups on the side chain, the final lignin product is called lignin sulfonate. Lignosulfonate and hemicellulose are dissolved in an aqueous solution, so the hemicellulose component should be removed first in the subsequent using process. Additionally, lignosulfonates have high molecular weight and high ash content [24]. All these characteristics make lignosulfonates suitable for many industrial applications, such as the production of adhesives, dispersants, surfactants, and others [27,28].

### 2.3. Soda Lignin

Soda lignin is isolated from the NaOH or NaOH/Anthraquinone pulping process, a method widely used in the treatment of non-wood raw materials such as rice, wheat straw, etc., as early as the 19th century [29]. As catalysts, anthraquinones can also attenuate carbohydrate degradation and promote lignin dissolution. Soda lignin and kraft lignin are collectively referred to as alkali lignin. In comparison to kraft lignin, soda lignin does not contain sulfur; therefore, it is more similar to natural lignin than kraft lignin and lignosulfonate, which also indicates that caustic soda lignin is more favorable for the chemical modification of lignin [30].

### 2.4. Organosolv Lignin

Organosolv lignin could be obtained by pulping with organic solvents such as acetic acid, formic acid, and ethanol. Compared with the above three kinds of lignin, the highlight feature of organosolv lignin is its relatively excellent solubility, which can be dissolved in alkaline solution and various organic solvents [26,31]. At the same time, organic solvent lignin has a relatively uniform molecular weight distribution due to the unique dissolution of the solvent, and it contains more condensation structures [23,31,32].

## 3. Techniques for Preparation of Lignin-Based Hydrogel

Generally, synthesis techniques of lignin hydrogel mainly include physical and chemical methods. Physical hydrogels are formed by physical interactions such as electrostatic interactions, hydrogen bonding, and chain entanglement. Chemical hydrogels are three-dimensional network polymers formed by crosslinking chemical bonds, and their properties are more stable than physical hydrogels. Among these, the functional groups that play an important role in the chemical reactivity of lignin are mainly phenolic hydroxyl groups, alcoholic hydroxyl groups, and carbonyl groups [33]. The aromatic ring of the lignin molecule is hydrophobic, and a hydrophilic group is introduced into the molecule to make it amphiphilic [34]. The modified lignin can be crosslinked with other monomers to prepare hydrogels.

### 3.1. Preparation of Lignin Hydrogels by Physical Crosslinking

Physically crosslinked lignin-based hydrogels are caused by physical interactions such as hydrogen bonds, electrostatic interactions, and van der Waals forces, making their formation process reversible. Additionally, the physical crosslinking method has the advantages of faster formation speed and less environmental impact. However, the hydrogel prepared by this method has poor stability, and the structure is easily destroyed under specific conditions (such as strong acid or strong alkali) [16]. The functional groups in the lignin building blocks are used to physically crosslink with other polymers via hydrogen bonding. The copolymers of hydroxyethyl cellulose and polyvinyl alcohol crosslinked using borax also produce hydrogen bonding interactions with lignin, and the resulting polymers show self-healing ability, which demonstrates the desired attributes [35]. Ionotropic gelation has also been found to form hydrogel via physical interaction. Ravishanker et al. developed a hydrogel using alkaline lignin and chitosan. Results illustrated that the formed hydrogel was attributed to the electrostatic interaction between the amine groups of chitosan and the phenolic hydroxyl anion of lignin [36]. The prepared hydrogel showed similar viscoelastic properties to chitosan-based gels but showed improved plasticizing properties due to the addition of alkaline lignin.

### 3.2. Interpenetrating Polymer Network and Polymerization Method

The interpenetrating network structure method refers to the introduction of lignin into the hydrogel structure in the form of interpenetration or semi-interpenetration. Lignin and other substances are independent of each other, and hydrogels with such structures have higher network density and stronger mechanical properties [37,38]. The fundamental mechanism is mainly due to free radical polymerization. The phenolic hydroxyl groups of lignin form free radicals in the presence of initiators, which react with monomer and/or polymer chains to form graft structures [39]. As shown in Figure 3a, the different monomers form a copolymer, in which the monomers react with lignin to generate lignin graft polymers through radical reaction. Afterward, the lignin-grafted polymer penetrates the network formed by the monomers, forming an interpenetrating polymer network (IPN) [40]. Xue et al. presented an acrylamide hydrogel using ethanol-organic solvent lignin as the reactive filler, *N*,*N*′-Methylenebisacrylamide as the crosslinking agent, and ammonium persulfate as the initiator [41]. The hydrogel had high water absorption, high flexural modulus, and excellent elongation at break, which could be used as a good water-retaining material. IPN strategy has also been used in the lignin–polyurethane hydrogel synthesis process, in which the lignin–polyurethane mixture was cast into the films and dried at 60 ℃ for 72 h. Further, the obtained films were hydrated with water and swelled to equilibrium in water. This prepared hydrogel had a double network structure with improved mechanical strength. In addition, the hydrogel showed superior compatibility with human dermal fibroblasts and was easily produced through 3D printing, casting into desired shapes, and fiber spinning techniques [42].

### 3.3. Crosslinking Copolymerization Method

Lignin-based hydrogels are prepared by direct crosslinking copolymerization of lignin or modified lignin and water-soluble polymers. Water-soluble high polymers mainly include polyurethane, polyethylene glycol diglycidyl ether, polyethylene glycol, and others. Hydrogels prepared by crosslinking copolymerization obtain a good swelling degree, thermal stability, and mechanical properties by regulating the content of lignin and are widely used in biosensing, flexible energy storage, and other fields [43,44]. Musilová et al. mixed glycine-modified kraft lignin and hyaluronic acid to obtain lignin–hyaluronic acid hydrogel with *N*-(3-dimethylaminopropyl)-*N*-ethylcarbodiimide hydrochloride as a crosslinking agent [45]. The hydrogel has good biocompatibility, swelling, and viscoelasticity, and the addition of lignin will not cause cytotoxicity to the hyaluronic acid hydrogel, which offers the possibility of application in tissue engineering and biomedicine. In addition, lignin and its derivatives react with acid catalysts and crosslinking agents such as formaldehyde and paraformaldehyde to form a network structure (Figure 3b). Due to the hydrophobicity of lignin, hydrophobic drugs are used as antibacterial materials and introduced into hydrogels through crosslinking [46,47]. Additionally, lignin is also developed to be mixed with inorganic materials to form organic-inorganic composites. For instance, nano iron tetroxide can be applied to the hydrogel to generate magnetic properties [48,49].

### 3.4. Graft Crosslinking Polymerization Method

The graft crosslinking polymerization method involves grafting monomers and other functional compounds onto the lignin building blocks, increasing the lignin’s reactivity. The grafted lignin is crosslinked with hydrophilic monomers through a crosslinker to obtain the different types of hydrogels. For example, a double bond is introduced into the lignin structure, which could further copolymerization. As shown in Figure 3c, the fundamental mechanism of this synthesis route is based on a radical reaction. Unsaturated segments are introduced into the lignin backbone through the esterification of phenolic hydroxyl groups of lignin to form unsaturated graft lignin. Further, this unsaturated lignin polymer is then copolymerized with other unsaturated monomers, such as hydroxyethyl acrylate, to form a hydrogel. The resulting lignin-based hydrogels are reported to have good water retention [50,51]. Based on this strategy, Ma et al. prepared grafted lignin through grafting *N*,*N*′-methylene-bisacrylamide to alkali lignin, which was further copolymerized with acrylic acid and other compounds (such as organo-montmorillonite) to fabricate the desired hydrogel with the initiator of ammonium persulfate (APS) [52]. In addition, Jin et al. introduced methacrylic acid groups to lignosulfonates. The synthesized methacrylic acid lignosulfonate (MLS) was used as a crosslinker to react with *N*-isopropylacrylamide and itaconic acid to prepare pH- and temperature-responsive hydrogels. The hydrogel had a temperature response at around 35 °C, which is close to the physiological temperature. The hydrogel also had a pH response when the pH value was 3.0 to 9.1, which has great application potential in the field of biomedicine [53].

## 4. Properties of Lignin-Based Hydrogel

### 4.1. Biocompatibility and Biodegradability

The biocompatibility makes lignin-based conductive hydrogel a suitable material to be applied in flexible electronic skin for detecting the pulse and heartbeat of the human body. The cell culture test and animal experiments are used to illustrate the non-toxic nature and skin compatibility of the hydrogel [54]. In addition, biodegradability means the lignin-based hydrogel will not pollute the environment and protects plants in the soil from microbes. The biodegradability of lignin-based hydrogels depends on the crosslinking density and the content of phenolics in the hydrogels [55]. The stronger the degree of crosslinking, the denser the pore structure of the gel, which reduces the accessibility to ligninolytic fungi and actinomycetes. Therefore, high crosslinking strength is more resistant to microbial attack than slightly crosslinked hydrogels. Phenolic substructures in hydrogels are directly attacked by enzyme systems expressed by most ligneous fungi; therefore, reducing the content of phenolic substructures in hydrogels could protect plants from fungal invasion [56,57].

### 4.2. Conductivity

In recent years, flexible conductive materials have shown potential applications in flexible energy storage devices, sensors, touch panels, and electronic skins, which have attracted more attention worldwide. The excellent flexibility and tunable mechanical properties of conductive hydrogels, combined with excellent electron and ion transport capabilities, have considerable development potential in the field of flexible electronic materials. Numerous synthetic strategies have been proposed to obtain various lignin-based conductive hydrogels, which are mainly conductive polymer-based conductive hydrogels, carbon-based conductive hydrogels, metal-based conductive hydrogels, and ionic conductive hydrogels [58].

Conductive polymer-based conductive hydrogels: Conductive polymers, as synthetic polymers, are characterized by their manner and ability to conduct electrons [59]. Currently, there are two main routes to prepare conductive polymer hydrogels. One is to gel the mixture of conductive polymers and hydrophilic polymers/monomers by self-assembly or introduction of cross-linkable elements. Another approach is to grow conducting polymers in prefabricated hydrogels by chemical oxidation and electrochemical polymerization. The electrical conductivity of conducting polymer hydrogels can be achieved by controlling the degree of polymerization of the polymers.

Carbon-Based Conductive Hydrogels: Carbon-based materials, such as carbon nanotubes, carbon fibers, graphene, and porous carbon, are considered to be suitable conductive materials for the preparation of conductive hydrogels due to their unique high electrical conductivity [60]. Among various carbon-based materials, carbon nanotubes and graphene are extensively studied in the field of materials science and are widely used as conductive fillers in conductive hydrogels for flexible and wearable electronics [61]. Carbon-based materials blended with various polymers and self-assembly methods after modification of carbon-based materials are two common methods for preparing carbon-based conductive hydrogels [62].

Metal-Based Conductive Hydrogels: Metals with low-dimensional nanostructures, such as nanowires, nanorods, and nanoparticles, are expected to be used in the preparation of flexible wearable electronic products due to their high electrical conductivity, catalytic properties, and easy preparation [63]. However, relatively few studies have been conducted on the preparation of hydrogels by direct addition of metal-based materials. Zhang et al. proposed the incorporation of sodium lignosulfonate into PVA hydrogels to form a microphase-separated structure and facilitate the adsorption of silver ions. A sodium citrate solution was further introduced as a green, reducing agent to reduce the silver ions adsorbed on the outer surface of the hydrogel. Thus, dense clusters of silver nanoparticles were formed on the surface of the hydrogel, and a conductive hydrogel with strong toughness was prepared [64].

Ionically Conductive Hydrogels: This hydrogel is composed of a three-dimensional framework structure and a continuous aqueous phase. Therefore, the hydrogel provides a large number of channels for ion migration, which is also an important property of ion-conducting hydrogel materials [65]. The ions are mainly derived from two components of the hydrogel (polymer network and solvent). The vast majority of polymer networks contain ionized groups (such as polyacidic monomers and acrylic acid) that generate freely mobile ions upon hydrolysis [66]. Polyzwitterions have both cationic and anionic charges on the same macromolecular chain, which ensures the high conductivity of hydrogels. Therefore, polyzwitterions also can be used to prepare ionically conductive hydrogels.

### 4.3. Porous Structure

Hydrogels have a crosslinked three-dimensional (3D) network structure, which can be divided into four types according to the shape and properties of the particles forming the network: a 3D network framework in which spherical particles are connected into chains and slowly formed, rod-like or lamellar structures are constructed as particles into a 3D network framework, cross-linked hydrogels composed of long macromolecular chains, in which some molecular chains in the hydrogel network framework are arranged to form microcrystalline regions, and gels formed by chemical crosslinking of long linear polymer chains.

For lignin-based hydrogels, adding a certain amount of lignin will increase the pore size of the hydrogel. However, when the lignin content exceeds the maximum value, the network structure is filled with lignin, and the pores in the hydrogel are closed [67]. The porous structure of the hydrogel is related to the lignin concentration. Hydrogels with lignin concentrations up to 5% (*w*/*w*) exhibit more ordered porous structures. However, the incorporation of lignin at concentrations higher than 5% (*w*/*w*) disorders the porous structure of the hydrogel, resulting in larger, more irregular pores and defects in the hydrogel texture [45].

### 4.4. Mechanical Properties

The mechanical properties of lignin-based hydrogels are affected by the kind and quantity of cross-linkers in the three-dimensional network structure of the hydrogel: (1) the network structure is formed by the molecular attraction between particles and particles. Such structures are often unstable because of the weak connection between the particles and are easily damaged under the action of external forces. (2) The network structure is constructed by relying on the role of hydrogen bonds. This type of hydrogel is more stable than the previous one. Moreover, the hydrogel contains more liquid, which provides it with elasticity. Molecular chains formed by hydrogen bonds can be partially parallel arranged into bundles to form locally ordered structures. (3) A network structure is formed in the hydrogel by chemical bonds between molecules. This type of hydrogel is very firm and stable, which has great application prospects in mechanics and thermodynamics. Generally, in the actual preparation of hydrogels, particle interactions may act synergistically and complement each other.

In addition, the mechanical properties (such as rheological properties, storage modulus, and loss modulus) of lignin-based conductive hydrogels are affected by lignin content. The increase in lignin content improves the mechanical properties of both the storage modulus (G′) and loss modulus (G″) of the hydrogel [68]. Due to the rigid network structure in the lignin hydrogels, the hydrogels exhibit higher G′ than the corresponding G″ in all selected angular frequency ranges [53]. In addition, the lignin content affects the mechanical strength properties in the lignocellulosic-derived hydrogel. The possible reason is that lignin molecules associated with each other form nano-aggregates, which make lignin precipitate and form a rigid phase in the hydrogel system [68].

### 4.5. Water Uptake and Retention

Polymer hydrogel materials refer to soft and wet materials with a low crosslinking degree that could quickly absorb and retain a large amount of water without dissolving in water. Similarly, lignin-based hydrogel is a class of rapidly developing functional polymer materials that integrate water uptake, water retention, and sustained release. There are two main reasons for the water uptake and water retention capacity of lignin hydrogel materials. One is the aforementioned porous structure of the lignin-based hydrogel. The swelling ratio is determined based on chemical groups such as hydroxyl and carboxyl groups and the size of the pores. Generally, larger pore size distributions are less dense, which correlate with higher swelling ratios [69]. The other is the functional groups in the lignin-based hydrogel. For instance, industrial lignin contains polar or hydrophilic groups such as hydroxyl, amino, carboxyl, sulfonate, etc.; thus, they can absorb tens of times or even thousands of times their volume of water and have a remarkable ability of water uptake [69,70].

## 5. Application of Industrial Lignin-Based Conductive Hydrogels

Generally, the common operating mechanism of conductive hydrogel for electronic devices is the external stimulus signals converted into electrical signals to realize the identification and detection of stimuli [71]. Sufficient conductivity, therefore, is essential for hydrogel electronic devices. In addition, stretchability, adhesion performance, bacteria, and UV resistance are also needed for the application of hydrogels in various fields [19]. Lignin contains abundant functional groups, which own excellent processability and reactivity. Meanwhile, lignin has great natural properties, such as biocompatibility, biodegradability, UV shielding, anti-oxidation, and antibacterial [19,72]. Industrial lignin, such as the aforementioned kraft lignin, soda lignin, lignosulfonate, and organosolv lignin, has been widely used in conductive hydrogels. In this section, we reviewed the most common applications of industrial lignin-based conductive hydrogels.

### 5.1. Bioelectronic Sensors

Due to the biocompatibility and biodegradability of lignin and lignin derivatives and the trait of hydrogel, lignin-based conductive hydrogels are sought as biomaterials for bio-electronic sensors to monitor heartbeat, pulse, human motion, and blood pressure.

Wang et al. developed a multifunctional organohydrogel sensor with poly (acrylic acid) (PAA) as the skeleton, poly (3,4-ethylenedioxythiophene): sulfonated lignin (PEDPT: SL) as conductive fillers, and water/glycerol as the dispersion medium (Figure 4a) [54]. The organohydrogel sensor showed conductive, self-wrinkled, soft and elastic, and anti-freezing properties, which sensed the movement of the limb, weak pulse, and throat vibrations. This conductive hydrogel could transmit physiological signals that were used for electromyography (EMG) and electrocardiography (ECG) detection. Additionally, cell culture test and animal experiments illustrated that the organohydrogel is non-toxic and could protect skin from frostbite.

Industrial lignin is also used as a catalyst for hydrogel-based bio-electronic sensors. For example, Fe-sulfonated lignin (SL)-g-polyacrylic acid (PAA) hydrogel/coating crosslinked with lignin-based nanoparticles-Fe^3+^ chelates was fabricated at room temperature, which showed high stretchable, self-healing, conductive, and UV-blocking properties (Figure 4b) [73]. It was discovered that the chemical reaction of SL-Fe^3+^ and ammonium persulfate (APS, initiator) generated semiquinone and hydroxyl radicals, which catalyze quick polymerization of acrylic acid monomers. Meanwhile, SL-Fe^3+^ chelates were used as crosslinking agents to impart the self-healing property of multifunctional hydrogels. Moreover, the gelation rate (1–5 min), mechanical, conductive, adhesive, and optical properties can be controlled by adjusting the ratio of SL and Fe^3+^. In addition, Zhang et al. also used the lignosulfonate (LS), Fe^3+^, and APS dynamic redox system to synthesize a double network hydrogel composed of lithium chloride, poly (vinyl alcohol), PAA, and LS. This hydrogel formed rapidly (in a few minutes) without other heating or UV radiation, which was used to detect human physiological activities, such as finger bending, knee bending, breathing, ECG, and EMG [74].

In recent studies, industrial lignin was also used as an adhesive agent for conductive hydrogel bio-sensors. For example, Wang et al. reported an LS-doped PAA hydrogel with high adhesive strength (30.5 KPa), stretchability (up to 2250%), conductivity (~0.45 S/m), and low compressive modulus (~20 kPa) (Figure 4c). The excellent adhesive performance of this hydrogel was due to the fact that the LS’s quinone groups formed physical crosslinking with PAA, and the catechol groups in LS interacted with various substrates by covalent linking, coordination bonds, hydrogen bonding, and π–π stacking. Moreover, the Fe^3+^ soaking strategy was used in this work to design an asymmetric adhesive hydrogel [75].

**Figure 4 polymers-14-03739-f004:**
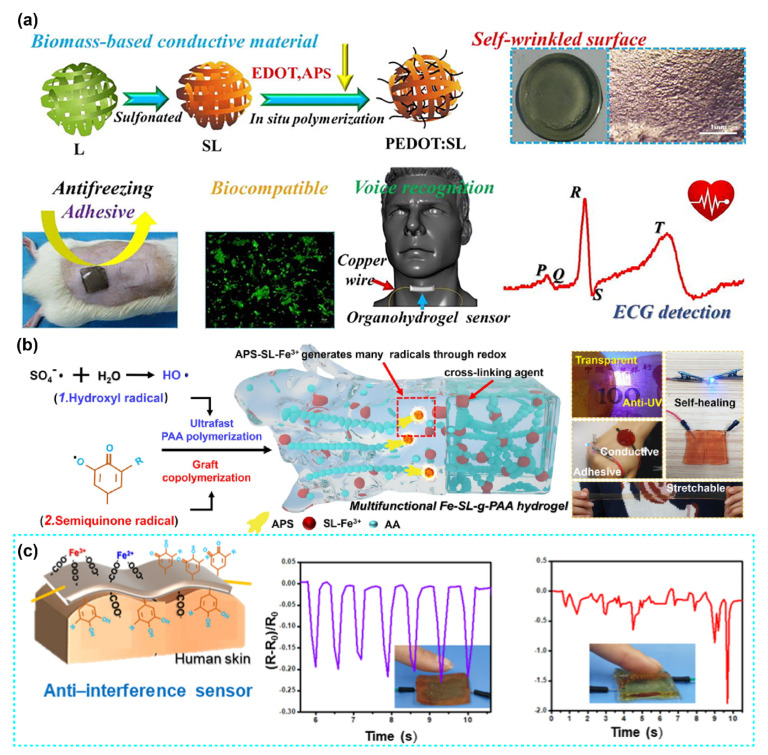
(**a**) Schematic for the PEDOT:SL-PAA organohydrogel synthesis, properties, and applications. Reproduced with permission from ref. [54]. Copyright 2019, Elsevier. (**b**) Schematic illustration of various free radical-initiated fast monomer polymerization and multifunctional hydrogels formed by SL/metal ion chelate crosslinked polymer chains. Reproduced with permission from ref. [73]. Copyright 2020, Elsevier. (**c**) Pressure sensors are assembled from asymmetrically bonded hydrogels. Pressure signals were recorded by Fe^3+^-treated P(AA-co-LS) and pristine P(AA-co-LS) hydrogel pressure sensors. Reproduced with permission from ref. [75]. Copyright 2021, Elsevier.

### 5.2. Strain and Pressure Sensor Devices

Benefiting from good biocompatibility and flexibility, hydrogel-based flexible strain and pressure sensors have attracted more and more attention in recent years and have high practical value in intelligent and electronic devices.

Han et al. developed a simple and low-cost method to prepare polyvinyl alcohol (PVA)/lignin–silver hybrid nanoparticles (Lig–Ag NPs) hydrogel-based piezoresistive sensors with superior mechanical performance and stable conductivity. The Lig–Ag NPs were formed due to the interaction between alkaline lignin and Ag NPs, which decreased the release of Ag NPs and generated stable semiquinone radicals in alkaline lignin. After compositing with PVA, Lig–Ag NPs provided a large amount of sacrificial hydrogen bonding and accelerated the delivery of electronic (Figure 5a). The obtained PVA/Lig–Ag NPs hydrogel presented outstanding pressure sensitivity, compressibility, and stability of signal response [76].

In another study, Cai et al. reported a conductive hydrogel with excellent mechanical strength and super toughness using enzymatic hydrolysis lignin (DEL) and biodegradable PVA as raw materials. DELs in PVA hydrogels can self-assemble to form nanophase-separated structures, which facilitate the adsorption of silver ions. After immersion in the green, reducing agent of sodium citrate solution, the silver ions in the hydrogel were reduced to form AgNPs. The silver nanoparticles in the hydrogel network not only endowed the hydrogel with good electrical conductivity but also enhanced the strength of the hydrogel by forming a metal-ligand coordination structure. The composite hydrogel also showed good electrical conductivity (about 1.0 S/m) and exhibited high strain sensitivity and stable response to deformation in compression, tension, and twist (Figure 5b) [77].

In addition, kraft lignin-based carbon (LC) as a conductive filler was used in the hydrogel system composed of PVA, cellulose nanofibrils (CNF), and carboxymethyl chitosan (CMC) to assemble pressure-sensitive hydrogel. The components of the hydrogel were crosslinked by hydrogen bonding among the carboxyl group, amino group, and hydroxyl group, which endowed the hydrogel with stretchability and fatigue resistance. Due to the presence of LC, this pressure-sensitive hydrogels exhibited sensitive deformation-dependent conductivity and could be used as flexible strain and pressure sensors (Figure 5c) [78].

### 5.3. Flexible Energy Storage

Lignin, as a renewable material, can produce energy, chemicals, and materials. The valorization of industrial lignin is considered a new path to reducing human reliance on non-renewable fossil fuel sources and exploring environmental feedstocks as functional materials. Lignin is a highly crosslinked aromatic polymer containing a large number of carbonyls and phenolic or phenolate structures [79,80]. Owing to these specific physiochemical properties, lignin is regarded as a promising candidate for the formation of high-performance electrodes and electrolytes.

Industrial lignin and lignin derivatives have been studied to synthesize gel electrolytes that can be used for flexible energy storage devices. For instance, a chemically crosslinked lignin hydrogel (SC lignin hydrogel) was prepared through catalytic ring-opening polymerization and crosslinking reaction. Additionally, a simple H_2_SO_4_ soaking strategy was used to convert the SC lignin hydrogel into a hybrid double-crosslinked lignin hydrogel (DC lignin hydrogel) by the formation of lignin hydrophobic aggregation (Figure 6a). The obtained DC lignin hydrogel presented outstanding shape recovery performance, high mechanical strength, and high ionic conductivity. The flexible supercapacitor assembled from this DC lignin hydrogel as the electrolyte and polyaniline-deposited carbon cloth as the electrode exhibited a high specific capacitance of 190 F/g, excellent energy density, and good cycling stability [81]. In another study, Park et al. developed a renewable flexible supercapacitor by combining chemically crosslinked alkali lignin hydrogel electrolytes with electrospun lignin/polyacrylonitrile nanofiber electrodes, as shown in Figure 6b. The lignin-based hydrogel electrolytes showed high ionic conductivity and mechanical robustness owing to their mechanical/dimensional stability and unique crosslinking chemistry. The lignin/polyacrylonitrile nanofiber electrodes with interconnected porous channels exhibited excellent charge storage capacity and kinetics. The supercapacitor device showed a high capacitance of 129.23 F/g and 95% capacitance retention over 10,000 cycles, as well as flexibility and durability at different bending angles [82].

In addition, lignin contains phenolic groups that can undergo redox reactions through the quinone/hydroquinone (Q/QH_2_) structure. Electrons and protons are stored in the Q/QH_2_ structure. The redox charge transfer of the Q/QH_2_ structure will effectively enhance the charge storage capacity if lignin is incorporated into the conducting electrode to allow charge transport [83]. Li et al. designed a novel mental-free and flexible supercapacitor based on a lignosulfonate functionalized graphene hydrogel (LS–GH) electrode and an H_2_SO_4_–PVA gel electrolyte [84]. Due to the reversible redox charge transfer of quinone groups, the electronic devices exhibited superior performance compared to other reported pseudocapacitive supercapacitors based on transition metal electrodes. The flexible solid-state supercapacitors based on an LS–GH electrode and H_2_SO_4_–PVA gel electrolyte demonstrated outstanding capacitive performance (408 F/g at 1 A/g) and good capacitance retention over 10,000 charge and discharge cycles (Figure 6c) [84]. A study showed that the carbonization of lignin alkali solution and adenine as nitrogen-doped activated carbon could be combined with graphene to obtain a hydrogel, which could be used as a supercapacitor electrode. The resulting flexible supercapacitor exhibited an energy density of 26.9 Wh/kg and excellent capacitance retention after 5000 cycles [85].

**Figure 6 polymers-14-03739-f006:**
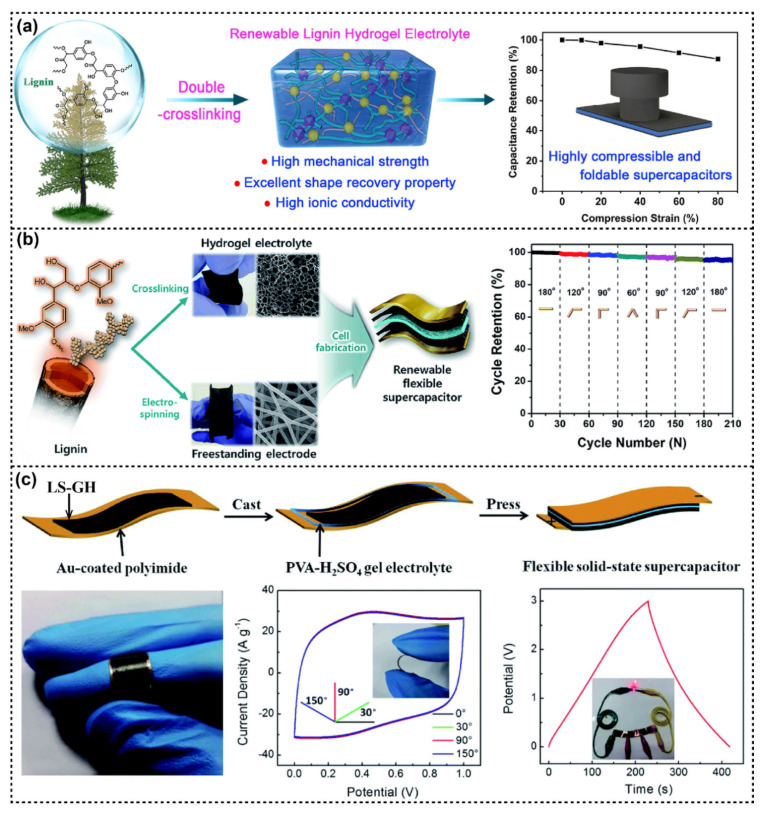
(**a**) Schematic illustration of a renewable double-crosslinked lignin hydrogel formed by sequential chemical crosslinking and physical crosslinking strategies, which could be an excellent hydrogel electrolyte for a highly compressible and foldable supercapacitor. Reproduced with permission from ref. [81]. Copyright 2020, Elsevier. (**b**) Experimental scheme for the fabrication of all-solid-state flexible supercapacitors based on lignin hydrogel electrolyte and lignin/PAN carbon nanofiber electrodes and cycling stability of lignin-based supercapacitor over 200 cycles under dynamic mechanical stress. Reproduced with permission from ref. [82]. Copyright 2017, Royal Society of Chemistry. (**c**) Schematic diagram of the fabrication process of the LS–GH flexible solid-state supercapacitor, digital photos of the flexible device, and electrochemical performance of the flexible device. Reproduced with permission from ref. [84]. Copyright 2019, Royal Society of Chemistry.

In addition to supercapacitor electrode and electrolyte materials, industrial lignin has also been used as a gel electrolyte in research on lithium batteries. For instance, Gong et al. fabricated a green and environmentally friendly gel polymer electrolyte based on the natural matrix of lignin, which showed higher conductivity, increased thermostability, and enhanced electrochemical stability compared to other common gel polymers. This study demonstrated the great potential of lignin as a gel polymer electrolyte in high-quality lithium-ion batteries [86].

### 5.4. Other Applications

In addition to the aforementioned applications, there are some other applications of lignin-based conductive hydrogels. For example, Hur and his coworkers developed a hybrid polypyrene (PPy)@lignin/agarose gel that could be used to detect the concentration of a target chemical analyte. The gel’s response to the added target analyte (such as ammonia) was recorded as a function of analyte concentration. The results illustrated that the PPy@lignin/agarose gel had the potential for simultaneous detection (Figure 7a) [87]. In another study, flexible carbon paste electrodes based on a functional glucose oxidase (GOx)/silica-lignin system were used to fabricate amperometric glucose biosensors. Lignin-functionalized SiO_2_ immobilized GOx on its surface. The GOx immobilization amount of biohybrid SiO_2_/Lig reached twice that of non-functionalized SiO_2_ [88].

Conductive lignin-based hydrogel has also been used in metal ions absorption. Sun et al. prepared a novel lignosulfonate-modified graphene hydrogel (LGH) to remove Cr^4+^ in an aqueous solution (Figure 7b). The results showed that the combination of lignosulfonate and graphene significantly improved the adsorption performance of LGH. LGH had a high specific surface area, interconnected porous structure, and a large number of oxygen-containing groups, which were especially suitable for the adsorption of heavy metals. The ultra-high adsorption capacity of the LGH for Cr^4+^ capture was 1743.9 mg/g [89]. In another study, Jiao et al. designed a novel sulfomethylated lignin-grafted-polyacrylic acid (SL-g-PAA) hydrogel. Moreover, the SL-g-PAA hydrogel (M^2+^@SL-g-PAA) was reused after adsorption to develop a chemiluminescence (CL) system (Figure 7c). The metal ions absorbed on SL-g-PAA hydrogel act as catalytic sites to catalyze the reaction between *N*-(4-aminobutyl)-*N*-ethylisoluminol (ABEI) and H_2_O_2_, which could increase CL intensity and long duration [90].

## 6. Conclusions and Perspectives

In summary, industrial lignin (kraft, lignosulfonate, soda, and organosolv) can be used as an additive to prepare composite hydrogels because of its low price, biocompatibility, low toxicity, eco-friendliness, and large number of polar oxygen-containing functional groups. Interpenetrating polymer network and polymerization, crosslinking copolymerization, and graft crosslinking polymerization are the three main methods for the preparation of lignin-based hydrogels. Emerging applications of lignin-based conductive hydrogels such as bioelectronic sensors, strain and pressure sensors, flexible energy storage devices, and metal/reagent detection sensors open up new avenues for industrial lignin valorization.

However, the currently reported lignin composite hydrogels have weak mechanical properties and limited functionality, which severely limit their popularization and practical applications. The main reasons are that most industrial lignin has an uncertain molecular weight and high polydispersity, contains more polar groups, and is prone to serious self-aggregation. The polar groups of lignin are encapsulated, resulting in poor interfacial compatibility between lignin and the hydrogel matrix. In addition, lignin has strong hydrogen bonding, which is prone to macroscopic phase separation during the preparation process, destroying the three-dimensional network structure of the hydrogel. Moreover, lignin-modified composite hydrogels generally have poor electrical conductivity due to the lack of an effective conductive medium.

Therefore, to improve the mechanical properties of the lignin-based hydrogel and further realize its conductive functionalization, future development of lignin in conductive hydrogel can be carried out in the following aspects according to the structure, source, and characteristics of lignin: (1) in the process of preparing the hydrogel, a combination of physical, chemical, microwave radiation, and other methods can be used to increase the active groups of the lignin and improve the reactivity of the lignin; (2) by modifying the phenolic hydroxyl groups on the surface of lignin or reducing the surface energy of the two-phase interface between lignin and hydrogel, the interaction between lignin and hydrogel is increased, thus enhancing the compatibility of lignin and other hydrogel materials; (3) appropriate physical, chemical, and biological methods should be selected to degrade and modify industrial lignin from different sources to improve the homogeneity of lignin molecular mass distribution and obtain functional groups with special functions.

## Figures and Tables

**Figure 1 polymers-14-03739-f001:**
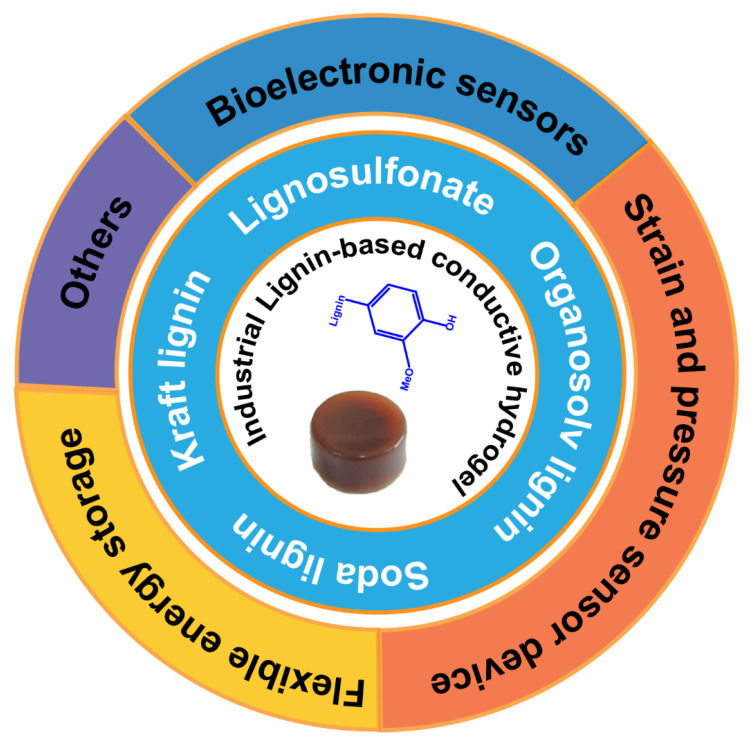
Schematic illustration of conductive hydrogels from different kinds of industrial lignin in different application fields.

**Figure 2 polymers-14-03739-f002:**
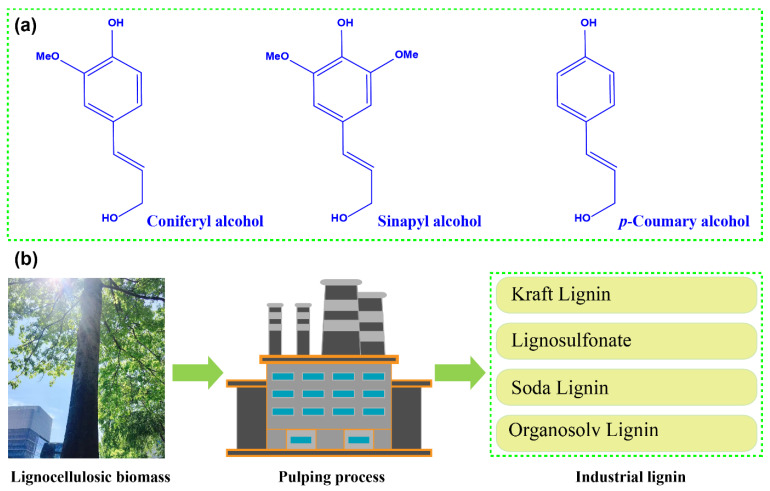
(**a**) Three monomers of industrial lignin. (**b**) Different pulping processes for the preparation of lignin derived from lignocellulosic biomass.

**Figure 3 polymers-14-03739-f003:**
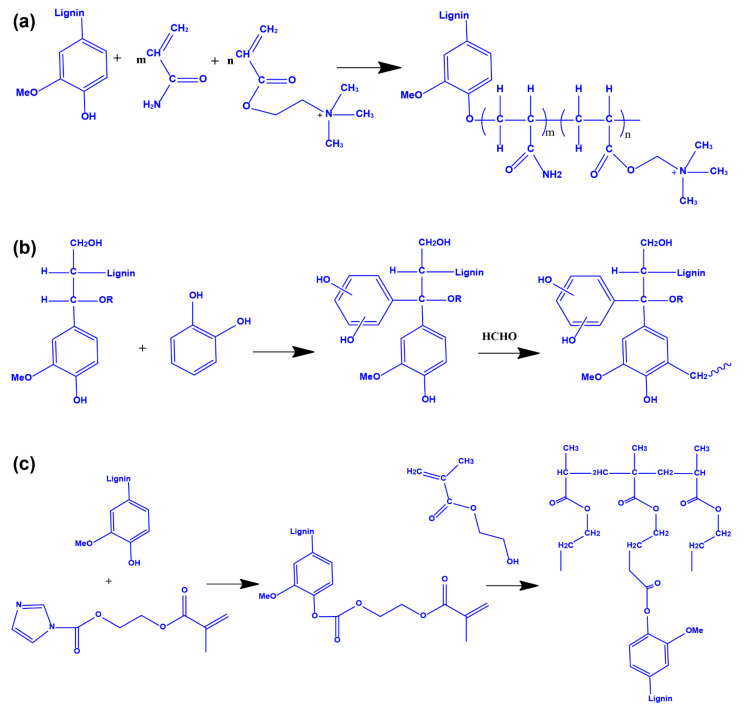
(**a**) Interpenetrating polymer network of lignin and crosslinking monomer. (**b**) Copolymerization of lignin and the crosslinking agent. (**c**) Copolymerization of grafted lignin and crosslinking monomer.

**Figure 5 polymers-14-03739-f005:**
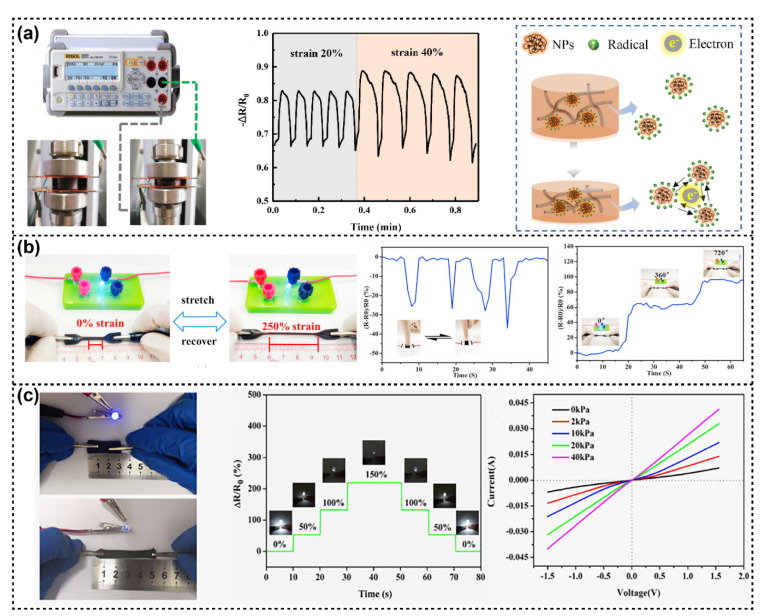
(**a**) Schematic diagram of the test setup, the resistive response of PVA/Lig–Ag hydrogels under repeated loading and unloading cycles at 20% and 40% compression, and the pressure-sensitive mechanism of PVA/Lig–Ag hydrogels. Reproduced with permission from ref. [76]. Copyright 2021, Elsevier. (**b**) The photograph shows the response of the LED to the applied strain of 0–250, real-time response to compression signals, and relative resistance changes of Ag@DEL torsion at 0°, 360°, and 720°, respectively. Reproduced with permission from ref. [77]. Copyright 2020, Elsevier. (**c**) Conductivity schematic diagram of PC/CNF/LC hydrogel pressure sensor, the responsivity of PC/CNF/LC hydrogel pressure sensor under different stretch lengths, current-voltage characteristics of PC/CNF/LC composite under different pressure. Reproduced with permission from ref. [78]. Copyright 2021, Elsevier.

**Figure 7 polymers-14-03739-f007:**
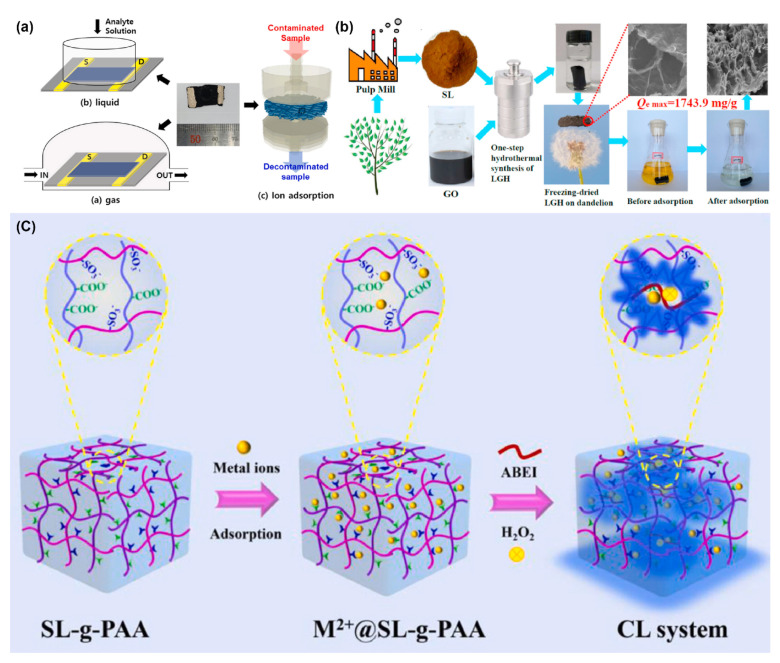
(**a**) Schematic illustration of the preparation of polypyrrole@lignin composites and their applications in electrical sensing and metal ion adsorption. Reproduced with permission from ref. [87]. Copyright 2022, Elsevier. (**b**) Schematic illustration of the synthesis of a novel lignosulfonate-modified graphene hydrogel and its ultra-high adsorption capacity for Cr (VI) in wastewater. Reproduced with permission from ref. [89]. Copyright 2021, Elsevier. (**c**) Schematic illustration of sequential application of lignin-based composite hydrogels for heavy metal ion adsorption and chemiluminescence. Reproduced with permission from ref. [90]. Copyright 2022, Elsevier.

## Data Availability

The data presented in this study are available on request from the corresponding author.

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
