# Peer review of "Conductive Hydrogels Based on Industrial Lignin: Opportunities and Challenges"

_polymers, 2022, doi:10.3390/polym14183739_

Round 1

Reviewer 1 Report

Comment to author:

This is an interesting manuscript that reviews the progress of conductive hydrogel using industrial lignin for biomedical application. They present interesting models for how conductive hydrogels and their fabrication methods could be elaborated to apply for biomedical application.  In particular, the use of conductive hydrogel in flexible electronic device and biomedical relevance could be attractive to the relevant fields including bioengineering, material engineering, and biomedical engineering. I recommend a minor revision for publication in Polymers. The manuscript could be also improved by attention to several matters:  

1) 4.1. Biocompatibility and biodegradability

“The biocompatibility made lignin-based conductive hydrogel a suitable material to be applied in flexible electronic skin, for instance, the detection of the pulse and heartbeat of the human body. “

- Do you mean “~ electronic skin sensor for detecting the pulse and heartbeat of the human body”?

2) 4.3. Porous structure

Hydrogels have a cross-linked three-dimensional network structure, which can be divided into four types according to the shape and properties of the particles forming the network. A three-dimensional network framework in which spherical particles are connected into chains and slowly formed; A three-dimensional network framework with rod-like or lamellar structures as particle structures; Three-dimensional network framework composed of long macromolecular chains, in which some molecular chains in the network framework are arranged to form microcrystalline regions; Gels formed by chemical cross-linking of long linear polymer chains.

- The  starting with “A three-dimensional ~ “ was used so many times.

3) “For lignin-based hydrogels, adding a certain amount of lignin will increase the pore size of the hydrogel. But when the lignin content exceeds the maximum value, the network structure is filled with lignin and the pores in the hydrogel are closed [68]. The porous structure of the hydrogel is related to the lignin concentration. Hydrogels with lignin concentrations up to 5% (w/w) exhibited more compact porous structures. But the incorporation of lignin at concentrations higher than 5% (w/w) disrupted the porous structure of the hydrogel, resulting in larger, more irregular pores and defects in the hydrogel texture [46].”

- The expressions ‘increasing’ and ‘compact’ are simultaneous in one paragraph. What is the point of the text?

4) 4.4. Mechanical properties

The mechanical properties of lignin-based hydrogels are affected by the connection properties of particles in the three-dimensional network structure of the hydrogel.

- The meaning of “particles” is ambiguous. (Other ‘particles’ in this article are likewise.)

(1) The network structure is formed by the molecular attraction between particles and particles. Such structures are often unstable because of the weak connection between the particles and are easily damaged under the action of external forces. (2) The network structure is constructed by relying on the role of hydrogen bonds. This type of hydrogel is more stable than the previous one. Moreover, because the hydrogel contains more liquid, which is elasticity. - correct English

 Molecular chains formed by hydrogen bonds can be partially parallel arranged into bundles to form locally ordered structures. (3) A network structure formed by chemical bonds. - It’s a phrase and correct English.

This type of hydrogel is very firm and stable, which has great application prospects in mechanics and thermodynamics. Generally, in the actual preparation of hydrogels, particle interactions may act synergistically and complement each other.

In addition, the mechanical properties (such as rheological properties, storage modulus, and loss modulus) of lignin-based conductive hydrogels are affected by lignin content. The increase of lignin content improved the mechanical properties of both the storage modulus (G') and loss modulus (G'') of the hydrogel [69]. Due to the rigid network structure in the lignin hydrogels, the hydrogels exhibited higher G' than the corresponding G'' in all selected angular frequency ranges [54]. In addition, the lignin content affects the mechanical strength properties in the lignocellulosic-derived hydrogel. The possible reason is that lignin molecule -> molecules associated with each other form nano-aggregates, which made lignin precipitate and form a rigid phase in the hydrogel system [69].

-> ‘made’ and ‘form’ need tense consistency

5) 4.5. Water uptake and retention

Polymer hydrogel materials refer to soft and wet material -> materials with a low cross-linking degree that could quickly absorb and retain a large amount of water without dissolving in water.

-In the 4.5 paragraph, you have used the past tense except for the first sentence. Please use the present tense.

- To describe the properties of the conductive hydrogel, are the properties of ‘biodegradability', ‘porosity’, and ‘water uptake’ necessary? They cannot support your theme.

6) You need to write the copyright of the figures correctly.

Reviewer 2 Report

The authors describe a review of conductive gels using lignin and provide a good summary of industrial lignin types and lignin-based hydrogels. However, the authors mention that flexibility and conductivity incompatibility are bottlenecks in biomaterial-based conductive hydrogels in the introduction section, but do not specify in the text how you are trying to solve this problem with lignin-based conductive hydrogels. And, The problems and their solutions to lignin-based conductive hydrogels are described in the second and subsequent paragraphs of the conclusion, but there is not enough data presented in the text to remind us of them. I request to add the above point. Also, please correct the following careless mistake

Page 7 to 9 at “Properties of lignin-based conductive hydrogel”

 Although the title of this chapter mentions conductivity, most of the descriptions are limited to the characteristics of lignin-based hydrogel, except for "Conductivity". How about changing the title of section 4 to “Properties of lignin-based hydrogel”?

Page15 at “Other applications”

 Despite the title of Section 5, Application of Industrial Lignin-Based Conductive Hydrogels, 91,92 cited in “Other applications” are papers that have nothing to do with conductivity. This is an inappropriate citation.

Page 11 Caption of Figure4.

The 3+ part of Fe3+ should be superscripted

Page11 2nd sentence 8th lines and 3rd sentence 12th lines

N for nitrogen in compound names should be italicized.

Round 2

Reviewer 1 Report

The manuscript is well revised and publishable.

Reviewer 2 Report

The authors have made sufficient corrections and the quality is good enough to be published in Polymers.